# A Sustainable Supply Chain Framework for Dairy Farming Operations: A System Dynamics Approach

Mohammad Shamsuddoha [1,*], Tasnuba Nasir [1] and Niamat Ullah Ibne Hossain [2]

1 Department of Management and Marketing, Western Illinois University, Macomb, IL 61455, USA
2 Department of Engineering Management, Arkansas State University, Jonesboro, AR 72401, USA
* Correspondence: m-shamsuddoha@wiu.edu

**Abstract:** The dairy industry plays a significant role in the global food system, providing essential nutrients for human consumption and creating rural employment. A small-scale dairy can assist a family in maintaining their livelihoods in Bangladesh. However, it is also associated with various environmental and social impacts, making it crucial for achieving sustainability. The triple bottom line of sustainability intends to achieve sustainability through improving productivity, implementing sustainable practices, and incorporating waste management. The dairy industry can continue to provide nutritious diets, ensuring sustainability practices. This research is a follow-up paper of Nasir et al. to find better sustainable results. It considers the triple bottom line of sustainability theory to improve the farm environment by reducing waste, managing resources efficiently, and promoting environmentally friendly practices. This paper is a case study on a dairy farm of 400 cattle in Bangladesh. The system dynamics method and simulation modeling were employed to draw dairy supply chain networks and examine the existing dataset to find better utilization of the dairy waste produced on the farm. Consequently, the simulation model incorporates waste management and value addition concepts to find better resource exploitation for gaining sustainable outcomes. Finally, this paper summarizes the simulation outcomes and articulates possible extensions for achieving further economic, social, and environmental benefits for the industry and surrounding community.

**Keywords:** dairy; sustainability; supply chain; system dynamics; Bangladesh

## 1. Introduction

The dairy industry in Bangladesh has experienced significant growth in recent years, with an increasing demand for dairy products and a growing number of dairy farmers. Despite this growth, the dairy farming sector in the country faces several challenges, including environmental degradation, low productivity, high input costs, breeding stagnation, disease, and adverse policies. A sustainable supply chain framework for dairy farming operations is crucial to address these challenges and ensure long-term sustainability [1–3]. Such a framework can help to ensure that the dairy farming industry is socially, economically, and environmentally sustainable. It can also enhance the efficiency and effectiveness of the supply chain, reduce waste and emissions, and promote the responsible use of resources. Sustainable practices positively impact infrastructure, education, and government policies. A diminutive research effort has been made in this context, whereas this particular sector dominates among the livestock businesses in Bangladesh. Using a system dynamics approach to develop a sustainable supply chain framework for dairy farming operations can provide valuable insights into the challenges being faced and potential solutions that can help to achieve sustainable growth. In this paper, we explore the development of a sustainable supply chain framework for dairy farming operations using a system dynamics approach. The study aims to identify appropriate operational mapping along with forward and backward supply chains to develop strategies to enhance sustainability. We begin with a case farm and its operational diagram and current data to model it in the simulation

environment to experiment with better fitting the available unused resources to obtain additional sustainable outputs. Finally, we present the results to establish sustainable outcomes through modified supply chains and operational processes.

The Brundtland Commission defines sustainability as 'meeting the needs of the present without compromising the ability of future generations to meet their own needs' (Brundtland Commission 1987 [4]). It encourages ideas of social impartiality, economic growth, and technological innovation while protecting the environment. In recent years, sustainability has been widely adopted by many individuals, organizations, and companies. A wide range of initiatives have been launched to encourage sustainability at various levels from individuals to corporations and have yielded positive results in reducing waste, conserving energy, and reducing greenhouse gas emissions [5–7]. It required a holistic approach to achieve positive outcomes for the environment, society, and the economy [8]. Developing sustainable practices and policies can result in improved ecological health, a more robust economy, higher-quality jobs, greater equality, and improved social welfare [9]. Again, the United Nations Sustainable Development Goals (SDGs) for 2015–2030 also provide a focused framework for creating sustainable development to balance economic, social, and environmental objectives [10]. They focus on poverty, education, health, climate change, economic growth, infrastructure, and industrialization [11]. They provide an important platform for governments, the private sector, civil society, and individuals to work together and engage with sustainable economic, social, and environmental developments [12]. These three aspects are related to the triple bottom line (TBL) which ultimately addresses the sustainability concern of an operation [13].

The Bangladeshi dairy industry faces a significant sustainability problem that has been exacerbated in recent years. The industry is overwhelmed by inefficient and unsustainable farming practices that result in low productivity and high resource use. This is further compounded due to the lack of proper waste management, which leads to significant environmental pollution. Furthermore, the issue of animal welfare is a growing concern, as many animals are kept in cramped and unhygienic conditions that can lead to illness and disease. The dairy industry in Bangladesh needs to adopt sustainable practices to ensure the long-term feasibility of the industry while mitigating its environmental impact. It is a critical factor for today and for generations to come through its long-term environmental and economic benefits and social implications for individuals, communities, and nations. In order to achieve more sustainable outcomes, this study is a follow-up of Nasir et al.'s 2013 paper and has two objectives:

- To develop and extend a dairy supply chain simulation model and fragment the model into economic, social, and environmental sustainability aspects.
- To examine the improvement of the economic gains, increase waste collection, utilize waste for producing valuable by-products, and increase social contributions through additional employment generation.

The ultimate goal is to enhance industry sustainability and positively impact the environment and society.

## 2. Materials and Methods

System dynamics is a business and social science approach to modeling complex system behavior over time [14]. It looks at the behavior of a system as a whole rather than focusing on its individual parts [15]. System dynamics allow one to explore how system structure and operations changes can affect performance [16]. In addition, system dynamics can be applied to any complex system, but it is beneficial for exploring the behavior of complex systems such as farming operations [17]. Similar methods are applied in the complex poultry supply chain to find sustainable outcomes [18,19]. Likewise, dairy farming involves many factors that interact differently and must be balanced to achieve the desired results. By applying system dynamics, one can better understand how these various factors interact and how to adjust them to optimize performance. For example, system dynamics can be used to evaluate the economic efficiency of dairy cattle production

systems [20]. Adjusting the feed ratio, grazing schedules, and reproduction planning can help identify the optimal efficiency for achieving maximum milk and meat production.

Various studies in the literature examine the impact of different breeding programs on the quality and quantity of milk produced. They also identify the ideal program for achieving the desired results [21]. System dynamics can help identify the optimal combination of cows and bulls to give the desired results [22]. A study was undertaken to develop a conceptual dairy supply chain model to examine sustainability [23]. This research uses the case of a dairy farm and its existing cattle, mature cow, milk production, and waste collection data. Such data help this study to build a simulation model and extend it in light of the triple bottom line theory. Enormous trials validate the existing data and then experiment with a few possible scenarios of economic, social, and environmental outputs to achieve sustainability. Several farm managers and supervisors were interviewed to cross-check the reliability, validity, and feasibility of the potential outcomes for validation. Overall, system dynamics can be an effective tool for optimizing dairy farming performance and creating a model of the various interconnected processes that impact the dairy industry, such as waste management, milk production, herd management, and financial gains. These analyses can help farmers make more informed decisions and optimize their operations for maximum profitability.

## 3. Bangladesh Dairy Sector

The dairy sector in Bangladesh plays a significant role in the economy due to its importance in providing food security, poverty alleviation, and income generation for a large proportion of its rural population [24,25]. Livestock farming, particularly dairy farming, has become an important livelihood for many farmers and the sector continues to proliferate throughout the country [26]. This sector has contributed significantly to economic development in the country and has been instrumental in providing food and nutrition security for its people. The FAO report [27] states that dairy is the largest agricultural sub-sector in terms of its direct contribution to gross domestic product, representing 7.75%. In terms of employment, it is estimated that 4.6 million people are directly employed in the livestock sector, including dairy, and another 10.3 million are indirectly employed in supporting services. With a 91% self-sufficiency rate, Bangladesh is currently ranked 25th in the world for milk production. Milk consumption has climbed by 13% over the past ten years, while milk production has increased by 18%. Despite the positive contributions, the dairy sector in Bangladesh has many areas in need of improvement. According to a World Bank report [28], only 35% of the milk produced in Bangladesh is marketed. The vast majority of the milk produced is consumed directly by households, meaning that the commercial milk sector is relatively small. This limits the investment potential and constrains the sector's potential growth and development. In addition, the current nutritional value of milk produced in Bangladesh is low due to a lack of appropriate feed.

The lack of appropriate storage facilities and infrastructure to connect rural dairy farmers with the market is also an essential constraint on the further development of the Bangladeshi dairy sector. The plan is a comprehensive set of initiatives to improve processing, animal health, safety and hygiene, and product quality and reduce wastage [29]. This increase in dairy production and consumption is mainly driven by improved access to dairy sector technology and increased access to inputs, such as high-quality feed and enhanced breeds of cows, as well as government support for the sector. The sector still faces numerous challenges. For instance, approximately 60% of dairy farms are small or unplanned operations, and the existing policy or formal sector does not serve 90% of them. There is also a large proportion of unorganized, unregulated sectors. Furthermore, there is a need for greater dairy sector training and capacity building. The current production costs are considered high due to many middlemen in the value chain and the high cost of inputs such as feed.

Ultimately, the sector has significant potential, but much work still needs to be done to maximize this potential. Increased access to credit, better training and extension services,

and the development of better value chains will be essential to unlocking the dairy sector's potential. The dairy sector in Bangladesh has a key role in economic development and in providing food and nutrition security to its people. However, many areas of improvement need to be addressed to realize their full potential. Government initiatives, strategic investments and adopting improved technologies and practices across the sector are essential to ensure the sector continues to grow and develop in the future. The above literature reveals a substantial number of challenges in the dairy industry. All these problems cannot be solved overnight but rather by step-by-step initiatives that will gradually resolve the issues. The priority is to bring the sustainability context to dairy operations, expectantly solving many of the above problems facing the farmers in general and the dairy industry in particular.

### 4. Sustainable Cattle Breeds in Bangladesh

The cattle breed is critical for the dairy business in which a farm intends to collect and raise sustainable breeds. Otherwise, obtaining enough milk or achieving meat productivity would make it hard to make the farm profitable. A farm needs to make an authentic inquiry to collect the right breed so that the farm will be sustainable in the long run. In Bangladesh, several dairy cattle breeds are commonly raised for milk production. According to Das, Islam et al., [16,30], Sahiwal, Red Chittagong, Hariana, and Gaolao are the common preferences of dairy farmers. The Sahiwal breed originates from the Sahiwal district of Pakistan and is known for its high milk production and adaptability to hot climates. Red Chittagong is a native breed known for its high resistance to tropical diseases and its good meat quality. On the other hand, the Hariana breed comes from India and is acknowledged for its high milk yield and adaptability to tropical climates. A few cows from the Gaolao breed are visible, which are renowned for increased milk production and adaptability to harsh conditions.

Determining the exact percentage of foreign cattle breeds in Bangladesh is difficult, as this information may not be consistently recorded or available. However, the Holstein-Friesian breed is estimated to make up a significant percentage of the total dairy cattle population in the country and is widely chosen by farmers. Several foreign breeds are available and somehow sustained in tropical weather. According to Samad [31], the most common foreign breeds in Bangladesh include Holstein Friesian, Jersey, and Brown Swiss, originating from the Netherlands, the British Channel Islands, and Switzerland, respectively. These breeds are sustained and chosen due to their high milk production and climate tolerance. The most sustainable dairy cattle breeds are well-adapted to the local climate and can thrive in the conditions found in the country. According to Samad [31], a few sustainable breeds in Bangladesh include Red Chittagong, Hariana, and Sahiwal. Red Chittagong is a native breed known for its high resistance to tropical diseases and good meat quality. Hariana and Sahiwal are well known for their high milk yield and adaptability to tropical climates. Hasan et al. [32] found that Sahiwal, Sindhi, Holstein Friesian, Jersey, Brahman, Red Chittagong Cattle, Pabna Cattle, and Mirkadim Cattle are also popular in so many dairy farms in Bangladesh. Thus, farmers need to ensure their collection of herds and followed with gradual genetic development for sustainable yielding to make their farms consistently profitable.

### 5. Dairy Sector and Sustainability

The Bangladeshi dairy sector is facing significant challenges in terms of sustainability, primarily due to persistent poverty, land fragmentation, lack of access to inputs, and inadequate government support [33]. On top of this, COVID-19 created more pressure on this sector to perform well as they failed to sell their milk in countrywide lockdown circumstances [34]. A key issue is the small size of landholdings and the reliance on traditional milk production systems where cows are allowed to graze rather than using pasture-based systems or intensification, which is common in many developed countries [35]. Most milk produced in Bangladesh is from traditional small-scale farms, leading to low yields

and limited effectiveness in utilizing the available resources and technologies. Several measures need to be undertaken to achieve a sustainable dairy sector in Bangladesh. These include providing access to improved production inputs, such as better-quality animal feed, improved livestock breeds, and veterinary care; creating incentives for farmers to adopt modern production systems; and introducing improved marketing and value chain management programs. There is also a need to improve access to credit and technical expertise and provide better access to land and pastureland. In addition, government policies and programs are prerequisites for improving farm productivity and environmental sustainability.

The dairy industry increasingly recognizes that it needs to become more sustainable to ensure its long-term success. Sustainability initiatives in the dairy industry include reducing water use, managing waste, improving energy efficiency, reducing emissions, and improving animal welfare. The dairy industry is also managing its waste more sustainably by using innovative technologies such as anaerobic digestion, composting, and biogas. Through anaerobic digestion, organic waste from dairy farms can be broken down and converted into methane. The methane produced is then used to generate energy; alternatively, it can be compressed and used as biofuel. This allows the dairy industry to cut its emissions and generate income from the sale of a renewable energy source. Dairies are also making their energy use more efficient. For example, by installing LED lighting, insulation, and efficient boilers and using renewable energy sources such as solar, wind, and biogas. These improvements can significantly lower their energy costs and help them to become more resource and energy efficient. The dairy industry also works to improve animal welfare by promoting better livestock management and welfare practices. This includes providing better ventilation and nutrition, reducing escape or injury hazards, and improving cow comfort. The dairy industry continues to strive toward sustainability, and with ongoing efforts and initiatives, the industry is becoming increasingly more sustainable.

In Bangladesh, the dairy industry is considered one of the most important sectors of agricultural production, contributing significantly to the national economy and providing employment to a million people [36]. However, despite this importance, the dairy industry in Bangladesh remains unsustainable with many challenges, such as inadequate infrastructure, poor quality inputs, and a lack of capacity for value addition. As a result, farmers are often unable to take full advantage of the potential of the industry, leading to limited income generation. Several initiatives have been launched to address this issue to enhance the sustainability of the dairy sector in Bangladesh. These include introducing improved breeds and breeds with higher milk yields, promoting pasture-based production, establishing linkages between farmers and processors, strengthening animal health services, adopting improved technologies, and promoting sustainable practices. Furthermore, public investment in infrastructure, access to credit, and inward investment from abroad have been identified as key drivers of sustainability.

Recent studies suggest that these efforts positively impact dairy sustainability. A study found that introducing new breeds, improved animal health services, linkages between farmers and processors, and increased access to credit had significantly increased income levels for smallholders within the sector [37]. The challenges faced by the dairy industry remain significant, but progress is being made. With the commitment of governments and other stakeholders toward further investment and policy reform, sectoral sustainability will likely improve in the near future.

## 6. Sustainable Development in the Dairy Sector

The rapid expansion of this industry has led to various environmental and ethical concerns, including greenhouse gas emissions, water pollution, and animal welfare. Implementing more environmentally friendly and ethical production processes is crucial to ensure the dairy industry's long-term sustainability. A critical aspect of sustainable dairy production is reducing greenhouse gas emissions. The production of milk is a significant contributor to global greenhouse gas emissions, with the majority of the emissions

coming from enteric fermentation (the release of methane from the digestive system of cows) and manure management. Various approaches have been proposed to reduce these emissions, including improving feed quality and management, reducing cow numbers, and developing new feed additives.

An approach that has shown promise is using high-quality forages and improved diets for dairy cattle. Forages, such as alfalfa, clover, and grass, can reduce emissions from enteric fermentation by improving the balance of nutrients in the cows' diets, leading to fewer emissions and less manure [38]. Additionally, feeding cows a diet that includes ingredients such as distillers' grains and citrus pulp has significantly reduced methane emissions. Another important aspect of sustainable dairy production is improving water management. In the past, dairy cattle were often confined and crowded together, leading to various health and welfare problems. However, increasing evidence suggests that more humane and natural production methods, such as pasture-based systems, can improve animal welfare and reduce the need for antibiotics and other treatments. By reducing greenhouse gas emissions, improving water management, and promoting animal welfare, we can ensure that the dairy sector continues providing essential nutrients and protein to millions worldwide while protecting the environment and promoting ethical production methods.

## 7. Achieving Sustainability in the Dairy Industry

Dairy Sustainability refers to the ability of the activity to continue indefinitely in a responsible manner. This includes protecting land, agricultural resources, and the economic livelihood of farmers. Achieving sustainability requires a multi-faceted approach, from adopting environmentally friendly farming practices and crop production systems to introducing government policies and incentivizing farmers to make farming more financially viable. Adopting sustainable agricultural practices is critical for the long-term sustainability of dairy farming. This includes using farm management systems with low environmental impacts, such as conservation agriculture, integrated crop-livestock systems, precision agriculture, and agroforestry. These systems have reduced environmental impacts over the long run, such as those related to soil, water, and air pollution. In addition, government policies should be introduced that protect smallholder farmers and promote the use of sustainable dairy farming practices. Policies should address access to land, economic incentives for farmers, credit policies, and public procurement schemes. Economic incentives should be provided to farmers to encourage the adoption of sustainable dairy farming practices. Government and private sector initiatives, such as price support policies, crop insurance, cost-sharing, and direct subsidies, can help farmers reduce risks and increase returns. Achieving dairy farming sustainability requires a multi-faceted approach, including the use of sustainable agricultural practices, government policies, and economic incentives.

There are several ways in which the dairy industry in Bangladesh can achieve sustainability; some of these include:

a. Improving animal health and welfare: Providing nutritious health care and housing for dairy animals can enhance their productivity and reduce disease-related costs.
b. Implementing efficient production systems: Adopting efficient production methods and technologies, such as modern milking systems, can reduce waste and increase output, making the industry more sustainable.
c. Promoting sustainable feed systems: Encouraging locally available feed resources such as grass and legumes can reduce feed costs and the industry's carbon footprint.
d. Encouraging waste management practices: Implementing waste management practices, such as composting and biogas production, can reduce environmental pollution and provide a source of energy for the industry.
e. Promoting sustainable water management: Ensuring that dairy farms have access to adequate and sustainable water supplies and implementing water-saving technologies, such as rainwater harvesting can reduce the environmental impact of the industry.

    f.    Encouraging community involvement: Building relationships with local communities and involving them in the decision-making processes of the dairy industry can increase their support for sustainable practices.

These are just a few examples of how the dairy industry in Bangladesh can achieve sustainability, as opined in various scholarly references and by farmers. To implement some of the above recommendations, the simulation model emphasizes the available resources and converts them into sustainable outcomes.

## 8. Dairy Supply Chain Model

This study uses a dairy industry case with 400 cattle including 200 growing heifers and around 200 milking cows. We used Nasir et al. [39] paper as a base paper to draw an initial model and extend it in light of the triple bottom line of sustainability aspects. The model (Figure 1) identifies preliminary operations based on economic, social, and environmental perspectives. From an economic perspective, the model considers heifers' cattle reproduction and successfully growing mature cows. Later, mature cows produce milk after successful cattle birth and in the following eleven months, they produce milk. Again, the employment generation considers a social contribution. Apart from this, more income will enable the farms to contribute towards a society that is an indirect input. The direct input will be employee welfare once the farm can have significant profitability. Most importantly, environmental contributions are visible through waste collection, followed by converting them into valuable by-products.

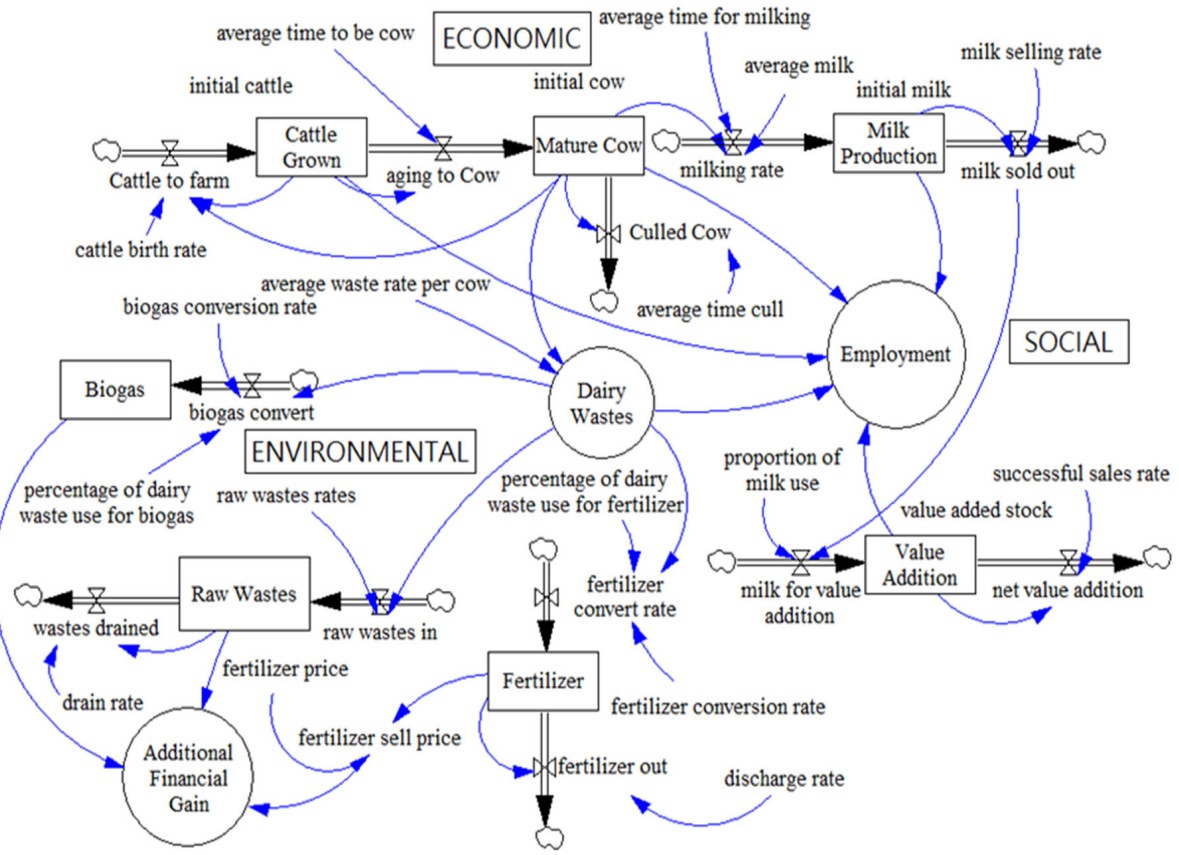

**Figure 1.** A simulation model for a dairy industry case.

## 9. Results from the Simulation Model

The simulation ran for 60 months (5 years) to examine the sustainable outcomes based on efficient waste collection and then reused them to convert value-added by-products.

Figure 2 depicts a gradual improvement in employment generation based on more milk production and value addition from the milk.

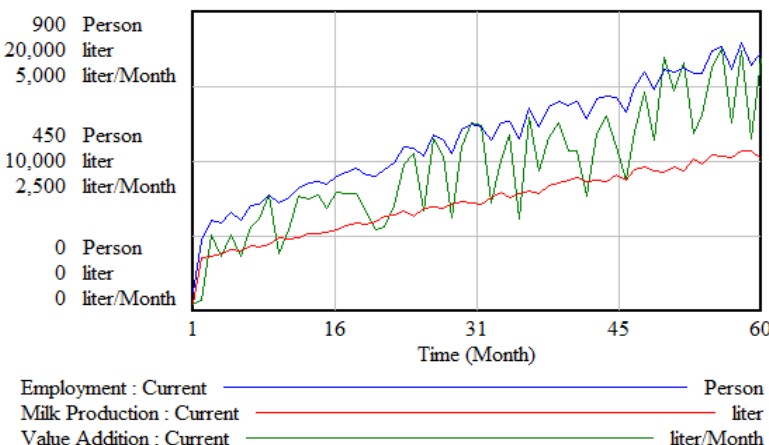

**Figure 2.** Simulation outputs on employment, milk production, and milk value addition.

Figure 3 represents the simulation outputs on cattle reproduction, the number of mature cows, and additional financial gains. It is noticeable that cattle reproduction is relatively consistent and mature cow numbers are growing sustainably. According to farm management, a healthy milking cow can reproduce 5–9 cattle and sustain more than ten years as a milking cow. This depends on the cow's health and resistance to various deadly diseases. Consequently, a greater number of mature cows will increase the number of milking cows and milk production. This model also considers the milk value addition of converting milk into sweet curd, sweets, ghee, butter, and dairy-related flavored drinks.

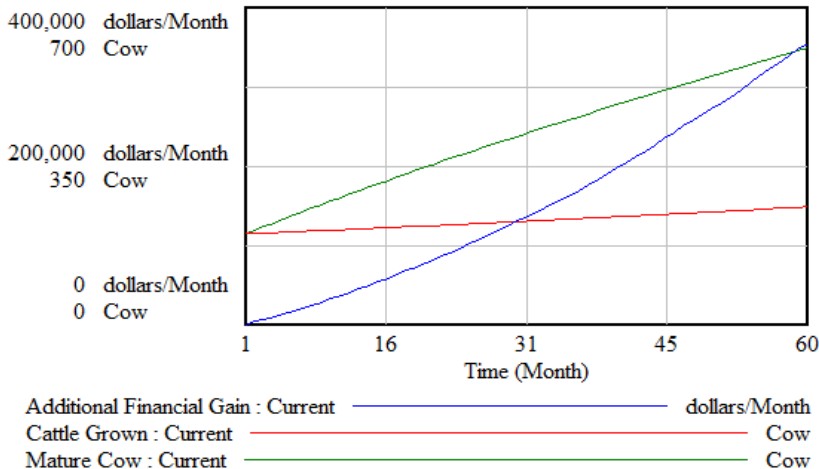

**Figure 3.** Simulation outputs on cattle reproduction, the number of mature cows, and additional financial gains.

Figure 4 represents the simulation outputs on fertilizer and biogas production from waste, which is crucial for dairy farms to eliminate to solve the surrounding environmental problems. Most dairy farms are accused of contaminating the surrounding environment by throwing waste into cultivable land, rivers, and canal water. This research especially considers this area and is devoted to collecting waste efficiently and converting it into valuable products of biogas and fertilizer and selling raw dung. The graphs show significant fertilizer and biogas conversions from unused waste. Such converted by-products can be sold on the market or self-consumed to save on utility bills. The below sections expose possible sustainable outcomes from the extended and experimented model outputs.

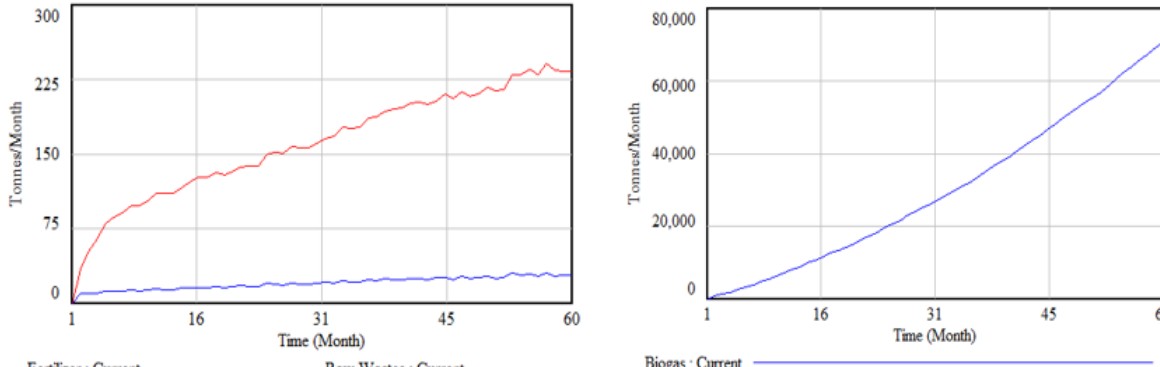

**Figure 4.** Simulation outputs on fertilizer and biogas production from waste.

## 10. Economic Benefits of Dairy Farming in Bangladesh

The simulation model ran for five years and explored futurist economic benefits, which are as follows:

1. Income generation: Dairy farming provides a source of income for farmers and their families, particularly in rural areas where other employment opportunities may be limited [40]. In addition to selling milk, dairy farmers can earn income from selling dairy products such as cheese, yogurt, and butter. Additional by-products can also be sold on the new market to gain further financial gains.
2. Employment creation: Dairy farming creates jobs for workers in producing and processing dairy products [41]. This can help to reduce poverty and unemployment in the country. Value-added products and waste management can accommodate more employment in this industry. In addition, more profitability will ensure that existing employees have better earnings and facilities.
3. Foreign exchange earnings: A country can export dairy products to other countries, generating foreign exchange earnings and contributing to economic growth [42]. The structural production system can maintain hygiene and compliance, leading to potential exports to other countries. The case farm exports its value-added products to various countries through third-party agencies.
4. Improved nutrition and food security: Dairy products provide a source of protein and other essential nutrients, contributing to improved nutrition and health for the population [43]. Dairy farming also helps to increase food security by increasing the supply of nutritious foods and reducing the dependence on imported food products. In this case, the farm ensures better food intake for the cattle, leading to nutritional milk products.
5. Rural development: Dairy farming can help to promote rural development by providing income and employment opportunities in rural areas, leading to improved living standards and infrastructure [44]. Such a structured rearing process and transparent supply chain help the farms to contribute to rural development through employment generation and selling by-products to the surrounding community. It creates more small businesses dealing with by-products.

## 11. Social Benefits of Dairy Farming in Bangladesh

In addition to its economic benefits, sustainable dairy farming also provides numerous social benefits, which are as follows:

1. Improved livelihoods: Dairy farming provides a source of income and employment opportunities, which can lead to improved livelihoods for farmers and their families [45]. The model outputs show relatively better income and employment to enhance the stakeholders' livelihood.
2. Better nutrition: Dairy products provide a source of protein and other essential nutrients, contributing to improved nutrition and health for the population (Huth et al. [46]).

3. Women's empowerment: Dairy farming provides opportunities for women to participate in the workforce, which can lead to increased economic independence and empowerment [47]), especially for rural women who deal with by-products such as raw dung, bio-fertilizer, and artificial charcoal from cow dung, and use biogas from the farm, ultimately empowering them effectively.

4. Increased access to new markets: Dairy farming can increase access to markets for farmers, enabling them to sell their products and obtain fair prices for their milk [41,48].

5. Rural development: Dairy farming can help to promote rural development by providing income and employment opportunities in rural areas, leading to improved living standards and infrastructure [23,49].

Therefore, dairy farming in Bangladesh plays a vital role in improving livelihoods and promoting social development in the country.

## 12. Environmental Benefits of Dairy Farming in Bangladesh

In addition to its social and economic benefits, dairy farming in Bangladesh also provides environmental benefits. Some of the key environmental benefits of dairy farming include:

1. Soil conservation and increased crop production: Dairy farming can help conserve soil by reducing soil erosion and promoting soil health using organic fertilizers and other sustainable farming practices [50]. The simulation model shows how to generate more and more fertilizers from the raw dung which will be a good source for surrounding agricultural land. Such bio-fertilizers can reduce chemical fertilizer usage, conserve soils, and increase crop production.

2. Biodiversity protection: Dairy farming can promote biodiversity by providing habitats for wildlife and preserving natural habitats through sustainable land use practices [51].

3. Climate change mitigation: Dairy farming can help mitigate climate change impacts by reducing greenhouse gas emissions through sustainable land use practices and using renewable energy sources in production processes [52]. For instance, the model shows how waste is converted into valuable by-products that help the farm reduce its carbon emissions.

4. Water conservation: Dairy farming can help conserve water by reducing water usage through efficient irrigation systems and promoting groundwater recharge through sustainable farming practices [23].

5. Waste management: Dairy farming can help to reduce waste by using sustainable waste management practices, such as composting and recycling, to reduce the amount of waste generated by production processes [53]. This is a focal part of the study and can resolve several problems simultaneously. There is plenty of scope there to improve waste management to restore the environment.

Thus, dairy farming in Bangladesh can provide numerous environmental benefits by promoting sustainable land use practices, reducing waste, conserving water and soil, and mitigating the impacts of climate change.

## 13. Triple Bottom Line Sustainability Successes in the Dairy Supply Chain Operations

The triple bottom line approach Elkington [54] considers a supply chain's economic, social, and environmental aspects in creating a sustainable and responsible system. To summarize the findings of the study, the following approach can be applied as follows:

1. Economic: The dairy industry has significant economic impacts, from milk production to selling value-added products. A sustainable supply chain should aim to increase profitability while ensuring fair and just compensation for all participants, including farmers, processors, and distributors.

2. Social: The dairy industry has social impacts, such as labor practices, food security, and community development. A sustainable supply chain should prioritize the well-being of all stakeholders, including workers, consumers, and local communities. This

can be achieved through fair labor practices, access to quality food, and investment in community development programs.

3. Environmental: The dairy industry has significant environmental impacts, from greenhouse gas emissions to land and water use. A sustainable supply chain should minimize its environmental footprint while promoting sustainable practices such as reducing waste, conserving natural resources, and mitigating climate change. To summarize, a sustainable supply chain in the dairy industry should balance economic, social, and environmental considerations to create long-term value for all stakeholders.

## 14. Contribution and Limitations

Theoretical contribution: There has been growing environmental concern about dairy farming operations in recent years. To address this, a sustainable dairy supply chain framework is required to integrate economic, social, and environmental factors, the circular economy, and reverse logistics. This study attempted to integrate outstanding theories through a system dynamics approach to develop an integrated, complex model to find positive outcomes. This approach can help producers and stakeholders identify critical areas for improvement and implement effective solutions that ensure the long-term sustainability of dairy farming operations. Thus, such theoretical advancements will help industry and academia to move forward to achieve more.

Practical contribution: This study modeled a dairy farm located in Bangladesh and considered existing supply chain networks to build further reverse logistics and circular economy extensions. Thus, this study can be implemented practically to be environmentally responsible in reducing waste and greenhouse gas emissions, minimizing water usage, and adopting efficient supply chain processes that improve milk production. The framework can also help improve farm profitability by identifying opportunities to convert waste into value-added products such as biogas, fertilizers, and required landfills. This approach can promote the long-term sustainability of the dairy farming industry by adding more value and having the same resources through practicing SD modeling.

Limitations of the study: This study narrowly focuses on a few variables for dairy farming operations, which limits our understanding of the total impacts on the industry and its stakeholders. The SD framework was developed based on a specific geographical context and a case farm which may not apply to different regions or contexts. Another limitation is the inherent complexity of supply chain dynamics, which may require additional research to fully understand the system's behavior and optimize its sustainability. Future research can fill the gap by adopting essential variables and examining the consequent effects on the total dairy supply chain and other processing networks to find optimality and sustainability using circular economy concepts.

## 15. Concluding Remarks

This paper extended the dairy supply chain networks by incorporating waste management loops along with redistributing them into different by-product sub-manufacturing units. The farm can efficiently collect and profitably convert most of its waste through this framework. Currently, farmers do not know how much waste they can expect from the feed intake ratio or cow herd size and age. Thus, the proposed supply chain framework will be handy for dairy farmers and the industry as a whole to estimate possible inputs and outputs. The changed simulation model and three sustainable aspects ensure better results than the current outputs. This model experimented with existing resources, and the model ran for the next five years and found excellent progress over time. However, for the industry to be completely sustainable, it is necessary to conduct more research investigations in the future. These include improving animal health and nutrition, enhancing additional milk production, improving genes and fodder management, increasing access to new local and foreign markets, adequate financing, and addressing further environmental issues such as unused biogas. A holistic approach involving the government, the private sector, NGOs, and farmers is required to overcome these challenges. The sustainability of

dairy farming also depends upon land and resource access, as many farmers struggle to obtain the land they need to raise and maintain their cows. Government regulations are necessary to ensure the sustainability of dairy farming. There should be laws in place to prevent the misuse of resources and pollution. Furthermore, these laws should incentivize dairy farmers to use sustainable practices. Shifts in consumer demand also have an impact on the sustainability of dairy farming. The demand for organic, hormone-free, and ethically sourced dairy products is on the rise. To keep up with this trend, dairy farmers must invest in organic practices, create organic labels, and provide full transparency of their products. Dairy farming can become a more sustainable industry by implementing sustainable strategies, introducing better government regulations, and keeping up with changing consumer preferences. With proper care and management, dairy farming can be a part of a thriving, sustainable rural community for years to come.

**Author Contributions:** Conceptualization, M.S. and T.N.; methodology, M.S. and T.N.; software, M.S. and T.N.; validation, M.S., T.N. and N.U.I.H.; formal analysis, M.S. and T.N.; investigation, M.S. and T.N.; resources, M.S., T.N. and N.U.I.H.; data curation, M.S. and T.N.; writing—original draft preparation, M.S. and T.N.; writing—review and editing, M.S., T.N. and N.U.I.H.; visualization, M.S., T.N. and N.U.I.H.; supervision, M.S. and N.U.I.H.; project administration, M.S. and N.U.I.H. All authors have read and agreed to the published version of the manuscript.

**Funding:** This research received no external funding.

**Institutional Review Board Statement:** Not applicable.

**Informed Consent Statement:** Not applicable.

**Data Availability Statement:** Will be provided upon request.

**Conflicts of Interest:** The authors declare no conflict of interest.

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
