# Peer review of "A Sustainable Supply Chain Framework for Dairy Farming Operations: A System Dynamics Approach"

_sustainability, doi:10.3390/su15108417_

Round 1
Reviewer 1 Report
A Sustainable Supply Chain Framework for Dairy Farming Operations: A System Dynamics Approach
The authors have targeted an interesting and trending concept of sustainable supply chain management. The manuscript can be improved to make it more suitable for publication in MDPI. Following are some of the comments to enhance this research.
1- The authors are requested to add a separate “Theoretical Implications” section before the concluding remarks.
2- Furthermore, the authors are requested to explain the “Practical Implications of the study,” followed by a section on theoretical implications.
3- The authors are further requested to add a separate section mentioning the “Limitations and future research directions of the study” after the section on practical implications.
4- The authors are requested to conduct a detailed proofreading check of the manuscript.
Thanks, and Good Luck
The authors are requested to conduct a detailed proofreading check of the manuscript.
Author Response
We appreciate your valuable comments, please see attached response file. Thank you.

Reviewer 2 Report
The reviewed article entitled “A Sustainable Supply Chain Framework for Dairy Farming Operations: A System Dynamics Approach” deals with a very important issue. The Dynamics Approach system was used in the manuscript. This approach is very adequate and appropriate. The authors conducted literature studies and correctly identified the research gap. The situation of the Dairy Sector in Bangladesh was also properly presented.
A very big advantage of the manuscript is the applied research approach - A System Dynamics Approach. On the other hand, the authors describe the assumptions of the model in very general terms. They indicate that data from a dairy farm was used. In addition, they inform that the model has been verified many times - also by "Several farm managers and supervisors". The authors do not provide information about the analyzed dairy farm. There is no basic information on the resources of production factors (land, labour, capital), type of production, etc. It is not known what activities were implemented, how dairy waste was used, how biogas was distributed, etc. The assumptions regarding the external environment are a separate issue. Did the authors try to simulate changes outside the farm - economic policy, weather, changes in prices of raw materials and products, etc. ?
Of course, knowing the basics of system modeling, I realize that it is practically impossible to describe all the mechanisms and assumptions (several dozen additional pages of a text). However, I recommend that the authors supplement the text with the basic information, e.g. on the dairy farm, and describe the assumptions regarding the simulation of changes in the external environment. I also think it is worth writing about the limitations of the model used. The research results seem to be very logical and consistent. The text is well-written and very interesting.
I think the manuscript is very valuable and should be published. However, I would be grateful if the Authors complete the missing information. Alternatively, this information can be provided in an annex.
Author Response

(The authors gave the same response as above.)
